behaviour/cognition/evolution

hired guns, alarm calls, *Cercopithecus nictitans*, recruitment

**Author for correspondence:**
Claudia Stephan
e-mail: cl.stephan@outlook.com

# Female putty-nosed monkeys (*Cercopithecus nictitans*) vocally recruit males for predator defence

Frederic Gnepa Mehon[1,2] and Claudia Stephan[1,2]

[1]Wildlife Conservation Society – Congo Program, Republic of Congo
[2]Nouabalé-Ndoki Foundation, Republic of Congo

CS, 0000-0001-6040-1701

Alarm calls can trigger very different behavioural changes in receivers and signallers might apply different alarm call strategies based on their individual cost-benefit ratio. These cost-benefit ratios can also vary as a function of sex. For instance, male but not female forest guenons possess loud alarms that serve warning and predator deterrence functions, but also intergroup spacing and male–male competition. In some forest guenons, the context specificity and alarm call repertoire size additionally differs between females and males but it remains unclear if this corresponds to similar sexual dimorphisms in alarm calling strategies. We here experimentally investigated whether general female and more context-specific male alarm calls in putty-nosed monkeys (*Cercopithecus nictitans*) had different effects on the opposite sex's behaviour and whether they might serve different female and male alarm calling strategies. We presented a leopard model separately to the females or to the male of several groups while ensuring that the opposite sex only heard alarm calls of target individuals. While female alarms led to the recruitment of males in the majority of cases, male alarms did not have a similar effect on female behaviour. Males further seem to vocally advertise their engagement in group defence with more unspecific alarms while approaching their group. Males switched alarm call types once they spotted the leopard model and started mobbing behaviour. Females only ceased to alarm call when males produced calls typically associated with anti-predator defence, but not when males produced unspecific alarm calls. Our results suggest that sexual dimorphisms in the context specificity of alarms most likely correspond to different alarm calling strategies in female and male putty-nosed monkeys.

# 1. Introduction

Alarm calls have been evolutionary sustainable because they change the behaviour of other individuals in ways that benefit the signaller. In social species, group members, including kin, usually adapt behavioural strategies to avoid predation upon receiving alarm calls [1,2,3]. Fitness benefits might be further increased if receivers could infer more specific information [4] that allowed them to apply predator-specific escape strategies [5]. In heterospecific associations, alarm calls can initiate cooperative, multi-species predator mobbing [6]. Towards predators that rely on stealth and ambush, alarm calls can also advertise their detection by potential prey, which often leads to the abandonment of predation attempts [7,8,9]. Hence, behavioural changes in others that are elicited by alarms can increase the inclusive fitness of signallers by either kin or individual selection. Additionally, alarm calling might also be linked to reproductive success. For instance, in multi-male baboon (*Papio cynocephalus ursinus*) groups, male alarm call rates are correlated with the dominance status of signallers, which probably mediates male–male competition over access to reproduction [10]. Hence, sexual selection may play an additional role in alarm call evolution, at least in species that form multi-male groups. The cost-benefit ratio that guides signaller behaviour can vary between individuals (e.g. depending on their spatial position and vulnerability, the number of present kin or opportunities for reproduction). If females and males systematically differ in costs-benefit ratios, this might lead to at least partly sex-specific strategies during alarm calling [11]. In these cases, sex-specific differences in alarm calls are expected to facilitate cognitive inferences from calls and behavioural interactions.

Some forest guenon species (*Cercopithecus* spp.) possess sexually dimorphic alarm calls with male and female alarms substantially differing in, e.g. call morphology [12,13], repertoire size [14] or in the distance over which their calls carry [15]. For instance, male forest guenons evolved laryngeal air sacs that allow them to produce high-intensity, loud alarms, a feature that is not present in females [16,17]. Male loud alarms are thought to serve two functions: (i) to warn group members about a present threat and to advertise predator detection, and (ii) to deter other groups and solitary males [18]. The usage of alarms that carry over considerable distances thus represents a male-specific communicative strategy. In several forest guenon species, sexual dimorphism is also present in the context specificity of alarms, with males possessing call repertoires that are to some degree indicative of the nature of ongoing events, while females use just one general alarm type [17]. This remains puzzling as alarm call repertoires, i.e. the specificity with which information about ongoing events is encoded, cannot be easily explained by differences in anatomy or vulnerability between females and males. Furthermore, forest guenons live in one-male groups in which males do not compete for reproductive access like seen in multi-male group species [10,19], although it would be generally difficult to explain how the number of different alarm call types could be subject to intra-sexual selection. Why did this sexual dimorphism in alarm call repertoires evolve in these species and do they correspond to different strategies in males and females during alarm calling?

Like other forest guenons, putty-nosed monkeys (*Cercopithecus nictitans*) live in one-male groups with various females and their offspring. Males usually socially interact less than females and they are often found at the group's periphery while females and offspring form a more cohesive cluster [20]. While females are philopatric, males disperse from their natal group and stay solitary or as part of heterospecific bachelor groups before they might take over a group. Although it is often females or younger individuals that spot a danger and start alarm calling, the male usually takes over a major role in risky, conspicuous anti-predator and territory defence [6]. By contrast, females and their offspring often try to hide in face of predators. Putty-nosed males often assemble their alarms into sequences that are to some extent indicative of different external events. Specifically, males usually emit 'pyows' in non-eagle related, terrestrial contexts including the presence of leopards (*Panthera pardus*), and 'hacks' to crowned hawk eagles (*Stephanoaetus coronatus*; electronic supplementary material S1 and S2). 'Pyow-hack' sequences are often associated with group movement [21]. Although call sequences usually start with calls matching the context of events, call types might be switched later during calling bouts [22]. Similar to blue monkeys [23], only pyows have been shown to convey reliable cues to signaller identity [24]. In the course of the present study, a third call type was consistently recorded towards a moving leopard model (electronic supplementary material S3). The call was named 'kek' and does not qualify as a loud call as it carries over considerably shorter distances than both loud calls, pyows and hacks (see the electronic supplementary material S7–S10 for details on acoustic parameter and cluster analysis results for male vocal repertoires). Females only possess one general alarm, 'chirp', that is emitted in response to all threats [22], figure 1; electronic supplementary material S4.

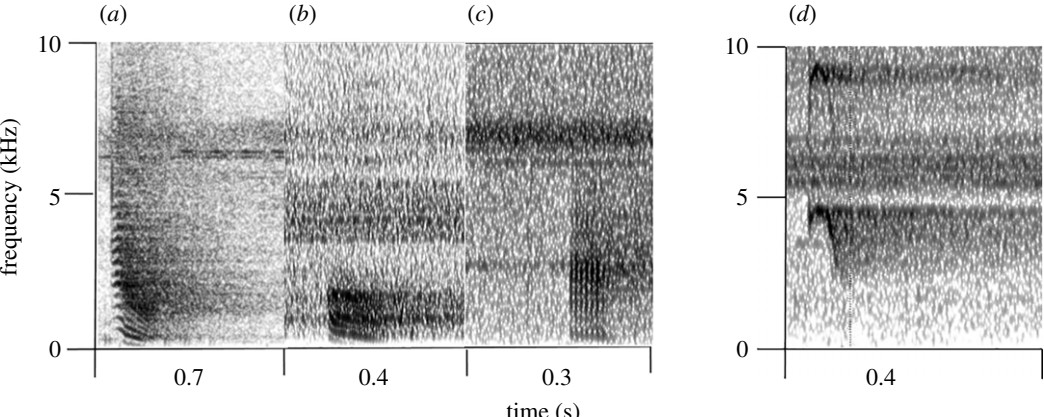

**Figure 1.** Spectrograms of (*a*–*c*) male and (*d*) female putty-nosed monkeys' alarm calls. Pyows (*a*) are usually uttered in response to various terrestrial disturbances while hacks (*b*) are closely associated with eagle presence. Keks (*c*) have not previously been reported and were emitted when males saw a leopard model. The calls are characterized by the absence of frequency transitions and are short in call duration (*n* = 234, median: 0.05 s, range: 0.04–0.06 s; electronic supplementary material, S6). In contrast with hacks, major bands of acoustic energy lay between frequencies of 0.5 to 3 kHz (in hacks: 0.6 to 1.1 kHz). Females possess a single alarm 'chirp' (*d*) that is uttered to all threats.

We here experimentally investigated communicative strategies in female and male putty-nosed monkeys during a simulated predatory event. Specifically, we separated the information about an ongoing event that was available to females and males. In the female condition, a leopard model was presented to females and males only heard female alarms. In the male condition, a leopard model was presented to the male of a group and now females only heard male alarms. We explored female and male call usage and behavioural interactions to see if response patterns were sex-specific. We predict that males are not able to infer specific information about the threat by only hearing general female calls and that they should approach and verify the situation themselves in order to adapt anti-predation strategies. For females, we predict no approaches to their calling male because more specific male alarms already provide them with information that allows for specific predation defence. We discuss possible alarm calling strategies and how differences in alarm call specificity could serve different strategies in females and males.

# 2. Methods

## 2.1. Subjects and site

We tested and observed 19 different groups of wild putty-nosed monkeys within 60 km$^2$ of mono-dominant *Gilbertiodendron dewevrei* forest in the Nouabalé-Ndoki National Park, northern Republic of Congo (2°15.50 N 16°24.70 E; altitude about 300 m) between January and June 2019, and in July 2020. Groups ranged from 7 to 21 individuals and consisted of one reproductive male, 3 to 11 adult females and their offspring (electronic supplementary material S5).

## 2.2. Experimental protocol

After the experimenter (F.G.M.) localized a target group in their known territory, group identity was confirmed based on specific group composition (electronic supplementary material S5) and characteristic features of specific individuals (e.g. a broken tail or scars). Females and the male were localized. Once it was confirmed that the male and the rest of the group were separated by at least 20 m (average distance: 40 m), the experimenter and an assistant separately monitored the females and the male with binoculars for 15–30 min to exclude disturbing events before trial onset and to keep track of individuals' position. Baseline vocal behaviour was then recorded for 3–5 min and, if the group was not moving, a second assistant covered himself in a leopard-printed fabric and approached either the male (male condition) or the majority of females (female condition). After the leopard model was spotted, it remained visible for 15 to 30 s. Females and the male were constantly monitored and their vocal as well as behavioural response was recorded until they ceased alarm

calling. The first assistant always stayed with the sex that did not see the leopard model. This was facilitated by species-typical, high degrees of female cohesion during resting and feeding. Dense vegetation did not allow for visual contact between females and the male, or for seeing the leopard model by the non-targeted sex.

A successful recruitment in the female condition (leopard model presented to females) was defined as the following: upon hearing female alarms, the male approached the rest of the group within the first minute after model presentation up to visual distance with females (behaviour reversed for sexes in the male condition).

None of the tested groups was habituated to human presence and each group is only represented once in the same condition. The order of conditions was reversed for 10 of the tested groups and trials of the different condition were separated by at least four weeks for the same group. Several trials had to be excluded from analysis owing to methodological reasons. These trials were repeated at most twice, separated by at least four weeks. If trials could still not enter analysis after the second repetition, the group was excluded for the respective condition (electronic supplementary material S6). This resulted in nine trials in the male condition and 17 trials in the female condition that entered analysis. Reasons to exclude trials include the following: alarm responses from neighbouring groups ($n = 4$), the experimenters had been detected ($n = 3$), crowned eagles appeared ($n = 3$) or the team had to retreat from forest elephants (*Loxodonta africana cyclotis*, $n = 3$) or gorillas (*Gorilla gorilla gorilla*, $n = 7$) during trials.

Vocal responses were recorded with a Sennheiser ME67 directional microphone and a Marantz PMD 660 solid-state recorder (44.1 kHz sampling rate, 16 bits amplitude resolution and stored in .wav format). Acoustic parameter for the cluster analysis (electronic supplementary material S7), call rates, female response duration and male alarm call types were extracted from these recordings using PRAAT 6.0.28 [25].

In July 2020, we observed the same groups to observationally assess whether females would usually approach their male or vice versa in non-predatory contexts, i.e. during feeding or after resting periods ($n = 24$ observations with five groups having being observed twice). If the majority of females and the male were at least 20 m apart, the observer stayed with the group and noted whether females or the male approached the other sex. In non-predatory contexts, the male of the group approached his females in 11 of 24 cases (46%) in contrast with 13 of 24 cases (54%), in which females approached their male. We thus set the expected proportion for female and male approaches to 0.5 and compared the proportion of sex-specific approaches in predatory contexts with this baseline.

## 2.3. Statistical analysis

Successful recruitments were compared for males and for females, using the binomial test with an expected distribution of 0.5, which was based on natural observations in non-predatory contexts.

Female and male relative call rates were calculated by dividing the number of calls by seconds of response duration. We compared female call rates (i) before and after the male's response onset (male condition), (ii) before and after the male's first alarm call (female condition) and (iii) before and after the male's first kek call (female condition) for each group. We opted for comparing the relative calling rate in each group using Wilcoxon signed-rank tests with an $\alpha$ level adjusted to 0.025 because of repeated testing of the same group during different response phases in the female condition. Male call rates were compared before and after females started alarm calling (female and male condition). When females were presented with the leopard model (female condition), we compared female response durations between trials with and without male kek calls using a Mann–Whitney $U$-test. For males, we were additionally interested in the call type with which males started their response, assuming that there is some link between call types used and the information that males inferred about the threat. Proportions of male responses starting with keks and with pyows, respectively, were compared between male and female trials using Fisher's exact tests.

Statistical analysis was conducted using R v. 3.6.1 [26]. All tests were two-sided.

# 3. Results

## 3.1. Recruitments

In the female condition, when females were confronted with the leopard model and started alarm calling, the male approached the rest of the group in 13 of 17 cases, which is more often than would be expected by usual male and female spatial behaviour (binominal test, $p = 0.041$). When the male reached his group,

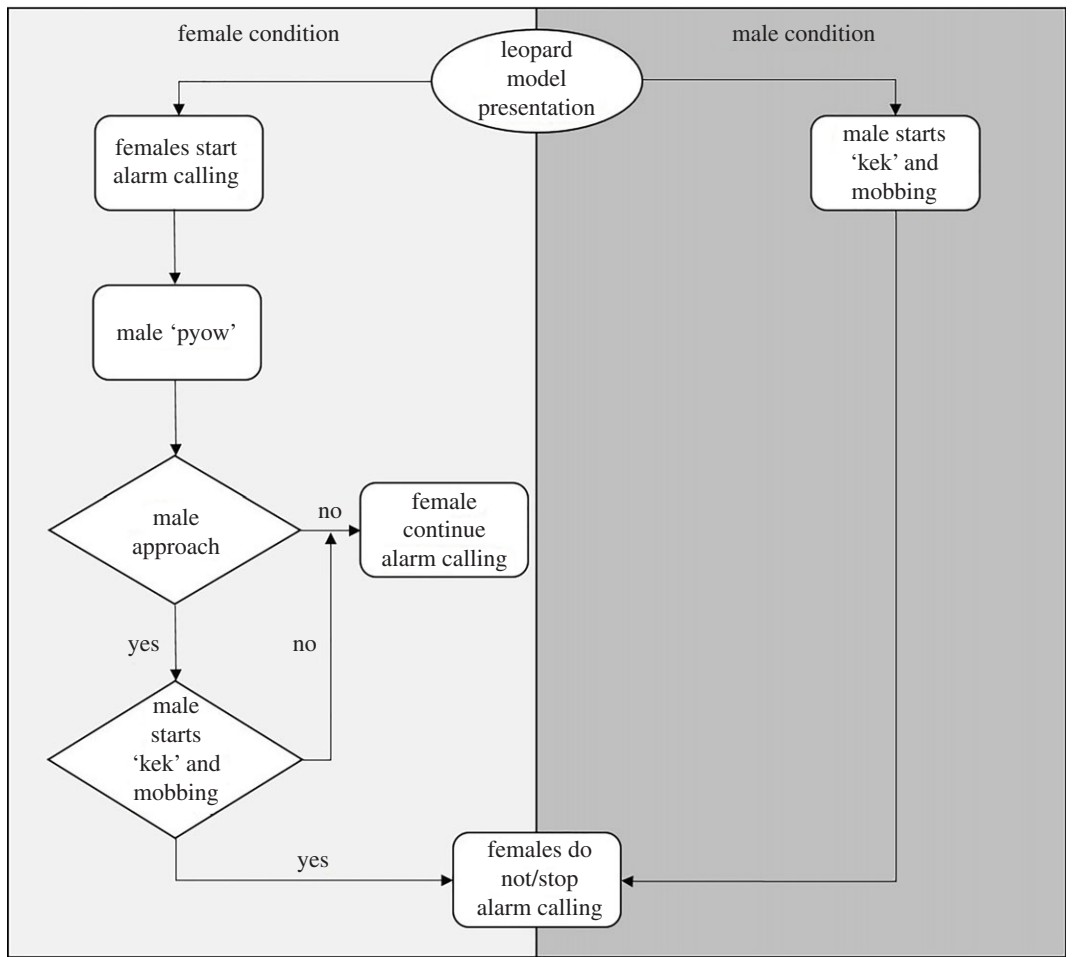

**Figure 2.** Male and female interactive response patterns to simulated leopard presence when females were presented with the model (female condition), or when the male of the group was presented with it (male condition). The circle represents the trigger (start point), boxes represent specific actions and diamond symbols encode decision points.

he scanned the surrounding area ($n = 13$). Upon detection of the leopard model, males descended from about 20 m to about 5 m towards the leopard model, showing typical mobbing behaviour like branch shaking (in 9 of 13 cases; figure 2).

In the male condition, when the male was confronted with the leopard model and started alarm calling, females never approached their male after the onset of his alarm response (0 of 9 trials; figure 2).

## 3.2. Vocal responses

In the female condition, when the male did not see the leopard model but only received female alarms, they were always triggered to emit pyow alarms ($n = 17$). When hearing their male's pyow calls, females did not change their alarm calling (before and after male pyows: $n = 17$, $W = 81$, $p = 0.586$; figure 3$a$). As soon as the male approached his group and detected the leopard model, he always switched from pyow to kek calls (9 of 9 trials). In response, females ceased their own alarm calling (before and after first male kek: Wilcoxon signed-rank test, $n = 9$, $W = 55$, $p = 0.003$; figure 3$b$). When the male did not approach females or he approached but did not switch to anti-predation defence (8 of 17 trials), the male also did not switch to kek calls and females persisted in alarm calling (female response duration: Mann–Whitney $U$-test, $n_{no\_kek} = 8$, $median_{no-kek} = 575$ s, $n_{kek} = 9$, $median_{kek} = 174$ s, $U = 8$, $p = 0.003$). Males called overall more when they approached their group than when they did not approach their group (successful recruitments: 0.46 calls s$^{-1}$ after female alarms, figure 3$c$; unsuccessful recruitments: 0.1 calls s$^{-1}$ after female alarms, figure 3$d$).

In the male condition, when males saw the leopard model first and females were oblivious to the nature of the threat, males uttered kek calls without preceding pyows and females hardly responded at all (before and after the onset of male responses: Wilcoxon signed-rank tests, $n = 9$, $W = 0$, $p = 0.99$;

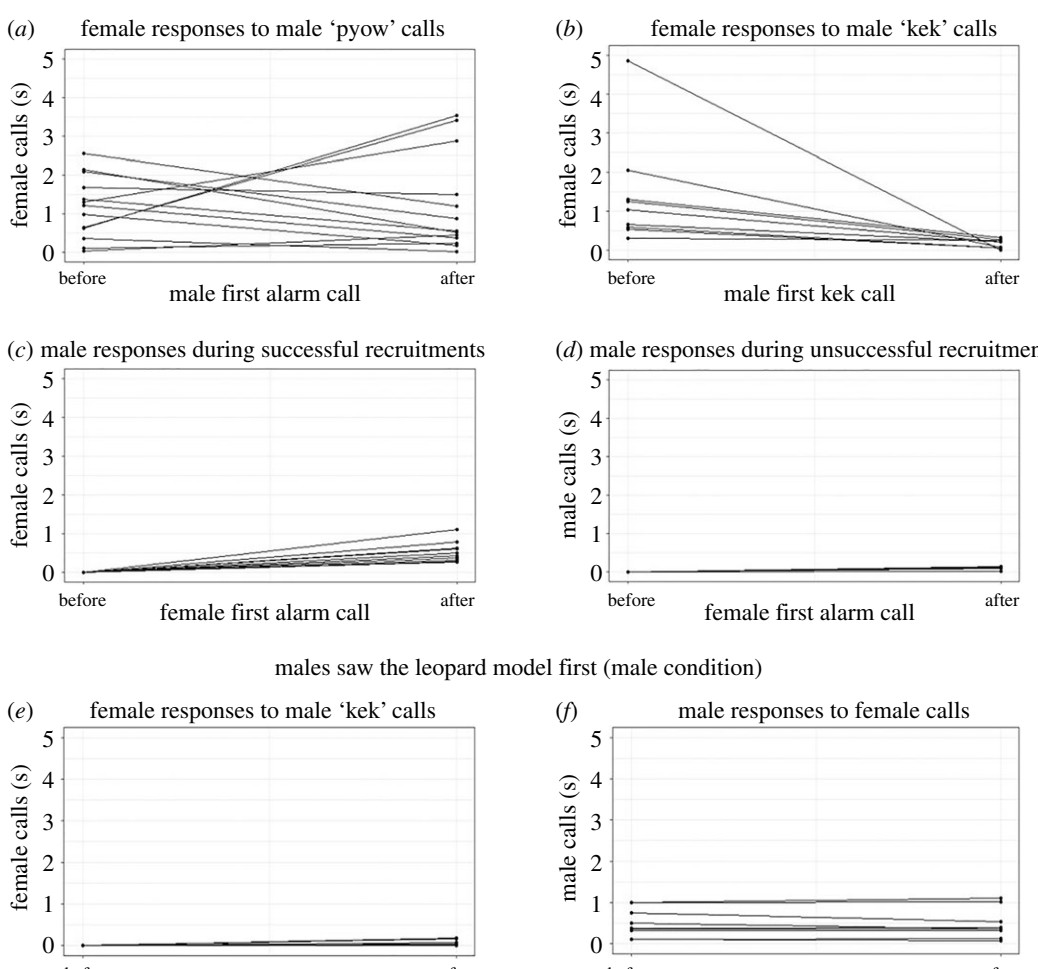

**Figure 3.** Relative call rates before and after calls from the opposite sex: (*a*) female alarm calls before and after the first pyow alarm in the female condition, (*b*) female alarm calls before and after the first kek alarm in the female condition, (*c*) male alarm calls before and after female first call in the female condition during successful recruitments, (*d*) male alarm calls before and after female first call in the female condition during unsuccessful recruitments, (*e*) female calls before and after the first kek call in the male condition, and (*f*) male calls before and after female alarms in the male condition.

figure 3*e*). Males did not change their call rates towards the leopard model after females started their weak alarm response (before and after the onset of female responses: $n = 9$, $W = 39.5$, $p = 0.96$, figure 3*f*).

When comparing male call type usage in female and male conditions, it followed consistent patterns. In the male condition, first male alarms were more likely kek calls (Fisher's exact test; the proportion of responses starting with keks: male condition versus female condition = 0.89 versus 0, $p < 0.0001$). By contrast, in the female condition, male first alarms were more likely pyow calls (proportion of responses starting with pyows: male condition versus female condition = 0.11 versus 1, $p < 0.0001$, electronic supplementary material S11b). Interestingly, male responses that did not contain any keks often contained hack calls (electronic supplementary material S11b), although no signs of crowned hawk eagles were spotted. Hacks were uttered alone and not as part of pyow-hack sequences.

## 4. Discussion

Our results show that female and male putty-nosed monkeys monitor each other's alarm calling and that their sex-specific alarm calls mediate behavioural changes in the opposite sex during simulated predation events. The elicited behavioural changes were different for males and for females, which leads us to assume that female and male communicative strategies are also sex-specific.

Specifically, our results promote the idea that female putty-nosed monkeys' general alarm requires males to assess the nature of the threat and that it serves to recruit males as 'hired guns' to ensure group defence. Accordingly, females were sensitive to different variants of male reliability for group defence, with females being more persistent in own alarm calling when males could not be recruited. This is in line with previous findings in female Diana monkeys (*Cercopithecus diana*), who are highly sensitive to their male's alarm call usage and who vocally persist if their male does not use his context-specific alarms according to females' assessment of an ongoing event [27]. Once the male takes over the anti-predation defence and distracts the predator, females can seek cover and hide vulnerable offspring. Female alarm call usage thus most likely evolved under individual and kin selection. Although male putty-nosed monkeys are easier to locate and thus more vulnerable than females when alarm calling, their alarm calling and engagement in predation defence is evolutionary beneficial because leopards usually interrupt predation attempts after they have been detected and because the only adult male is likely to have sired most offspring in the group. Thus, individual and kin selection also explains alarm calling in males.

One phenomenon remains puzzling though. Why should males call while approaching their group and why should they use two different call types? Given that males never recruited females for collaborative group defence, we suggest that males advertise their engagement in predation defence with more general pyow calls while approaching the rest of their group without yet having more detailed information about the nature of the threat. Kek calls were only uttered when males detected the leopard model and started predator mobbing. Male alarm calling was indicative of male behavioural engagements in predation defence, with males calling overall less in trials in which they did also not approach the group after receiving female alarms. Females seem to infer male behaviour from pyow and kek calls and only immediately ceased own alarm calling when receiving male kek calls.

The kek calls, firstly reported in the current study, and mobbing towards a visual, moving leopard model were closely linked. A previous study on putty-nosed monkeys that used a static leopard model did not report on this call type [28]. Whether these calls are population-specific and whether they are only uttered towards moving threats on the ground, provides a promising topic for future studies.

In summary, results presented here show that female and male putty-nosed monkeys follow different strategies when alarm calling that can be partly explained by individual and kin selection. However, individual and kin selection do not straightforwardly explain why males possess different call types and why they start calling during approaches without having information about the threat. The fact that pyow calls provide females with information about the signaller's identity [24] could be important for the group-holding males' reputation as reliable group defenders and their tenure track length [29], especially in dense forest habitats where visibility is usually low. It yet remains unknown to what degree female forest guenons have control over mate choice as seen in non-human primates forming multi-male groups [19,30,31]. One specific hypothesis would be that males, who do not reliably take over anti-predation defence also have less reproductive success, by e.g. shorter tenure duration, than more reliable callers and group defenders. The investigation of male reliability in group defence as indicated by alarm call usage, and female mate preference provide promising ground for future research.

Ethics. We are grateful to the Ministère de l' Economie Forestière, the Agence Congolaise de la Faune et des Aires Protégées (ACFAP), and to the Institut en Recherche Forestière within the Ministère de la Recherche Scientifique et de l'Innovation Technologique for permission to work in the Nouabalé-Ndoki National Park (research permit no. 270/2020, 001/2021).

Data Availability. The data analysed here are provided as the electronic supplementary material.

Authors' contributions. F.G.M. and C.S. designed the study. F.G.M. collected the data. F.G.M. and C.S. analysed the data. C.S. wrote the first draft. F.G.M. and C.S. revised the first draft and approved the final version.

Competing interests. We declare we have no competing interests.

Funding. The current research received funding from US Fish and Wildlife Service, the European Union, USAID's Central Africa Forest Ecosystems Conservation Project, Fondation Tri-National Sangha, Columbus Zoo and Aquarium, Cincinnati Zoo and Botanical Garden, Woodland Park Zoo, Riverbanks Zoo and Garden, Dublin Zoo, Dutch Gorilla Foundation, and Koeln Zoo.

Acknowledgements. We are grateful to the Ministère de l'Economie Forestière, the Agence Congolaise de la Faune et des Aires Protégées (ACFAP), and to the Institut en Recherche Forestière within the Ministère de la Recherche Scientifique et de l'Innovation Technologique for permission to work in the Nouabalé-Ndoki National Park (research permit no. 270/2020, 001/2021). Particular thanks go to Jean-Pierre Peya, Alain Bimba, Lazard Libanga and Mathias Ekoutou for invaluable help and protection during data collection. We are grateful to Sean Brogan for proof reading English language and to him as well as to three anonymous reviewers and two editors for comments that clearly helped to improve a previous version of the manuscript.

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
