## [Peer Review File · Royal Society Open Science]

Review History

RSOS-202135.R0 (Original submission)

Review form: Reviewer 1

Is the manuscript scientifically sound in its present form?

Yes

Are the interpretations and conclusions justified by the results?

No

Is the language acceptable?

No

Do you have any ethical concerns with this paper?

No

Have you any concerns about statistical analyses in this paper?

Yes

Recommendation?

Major revision is needed (please make suggestions in comments)

Comments to the Author(s)

See enclosed file (Appendix A).

Decision letter (RSOS-202135.R0)

Dear Dr Stephan

The Editors assigned to your paper RSOS-202135 "Female putty-nosed monkeys (*Cercopithecus nictitans*) vocally recruit males for predator defence" have now received comments from reviewers and would like you to revise the paper in accordance with the reviewer comments and any comments from the Editors. Please note this decision does not guarantee eventual acceptance.

Please submit your revised manuscript and required files (see below) no later than 21 days from today's (ie 25-Jan-2021) date. Note: the ScholarOne system will 'lock' if submission of the revision is attempted 21 or more days after the deadline. If you do not think you will be able to meet this deadline please contact the editorial office immediately.

on behalf of Dr Oliver Schülke (Associate Editor) and Kevin Padian (Subject Editor)
openscience@royalsociety.org

Associate Editor Comments to Author (Dr Oliver Schülke):

Dear Dr. Stephan,

As the associate editor handling your submission, I have now received comments on your revised version from one of the previous reviewers. Based on these comments, I will suggest to my editor to ask for another round of major revisions. Please follow the suggestions of the reviewer carefully, consider again whether you are not over-interpreting results with respect to sexual selection theory, and include the changed text passages in your response letter.

I am looking forward to receiving a revised version of the manuscript at your earliest convenience.

With kind regards,
Oliver Schülke

Reviewer comments to Author:

Reviewer: 1
Comments to the Author(s)

See enclosed file

===PREPARING YOUR MANUSCRIPT===

Your revised paper should include the changes requested by the referees and Editors of your manuscript. You should provide two versions of this manuscript and both versions must be provided in an editable format:
one version identifying all the changes that have been made (for instance, in coloured highlight, in bold text, or tracked changes);
a 'clean' version of the new manuscript that incorporates the changes made, but does not highlight them. This version will be used for typesetting if your manuscript is accepted.
Please ensure that any equations included in the paper are editable text and not embedded images.

If you have been asked to revise the written English in your submission as a condition of publication, you must do so, and you are expected to provide evidence that you have received language editing support. The journal would prefer that you use a professional language editing service and provide a certificate of editing, but a signed letter from a colleague who is a native

speaker of English is acceptable. Note the journal has arranged a number of discounts for authors using professional language editing services (<https://royalsociety.org/journals/authors/benefits/language-editing/>).

===PREPARING YOUR REVISION IN SCHOLARONE===

<https://royalsociety.org/journals/authors/author-guidelines/#supplementary-material> to include a suitable title and informative caption. An example of appropriate titling and captioning may be found at https://figshare.com/articles/Table_S2_from_ls_there_a_trade-

off_between_peak_performance_and_performance_breadth_across_temperatures_for_aerobic_sc
ope_in_teleost_fishes_/3843624.

Author's Response to Decision Letter for (RSOS-202135.R0)

See Appendix B.

Decision letter (RSOS-202135.R1)

Dear Dr Stephan,

It is a pleasure to accept your manuscript entitled "Female putty-nosed monkeys (*Cercopithecus nictitans*) vocally recruit males for predator defence" in its current form for publication in Royal Society Open Science.

on behalf of Dr Oliver Schülke (Associate Editor) and Kevin Padian (Subject Editor)
openscience@royalsociety.org

Appendix A

Review of „Female putty-nosed monkeys vocally recruit males ...“

The authors have put considerable effort into the revision of the manuscript, but several issues remain. Before I go into the details, I would like to encourage the authors – should they consider to revise the paper – to paste the changed sections as detailed responses into the response letter. Simply saying that one has changed the text and then referring to the line in the text makes both the editor’s and the reviewers’ job unnecessarily hard.

Overall, there are some major improvements, but the paper is hard to follow, and there are still some passages where the structure could be improved. The discussion is not to the point. I strongly recommend a careful revision, perhaps with the aid of an experienced colleague to make the paper more accessible. The data in itself are valuable and could be useful for developing a more fundamental understanding of the factors that shape the structure and usage of alarm calls.

Specific comments:

The title only captures half of the story.

Abstract: Overall, the abstract is poorly structured. The first sentence is about call structure, the second about call usage and responses to calls.

L 14: “Nonetheless, communicative strategies ... “ This statement appears to be a straw man. Why don’t you start with a more general statement that the costs and benefits of alarm calling are expected to vary in relation to individual vulnerability, the presence of kin in the group, and sexual selection. These factors may explain why alarm call patterns and responses to alarm calls may differ between the sexes. Then you go on with “We here experimentally investigate”. The thought outlined above could be unpacked in the introduction section.

16: “another forest guenon” – the reader can only guess what you are trying to say. Delete “another”

19: reassuring → replace with “ensuring”

22: delete “had”

23: Full stop after “group”, then start new sentence with “They ...(and insert “also” here).

25: Change to “females only ceased to alarm call when males produced calls typically associated with anti-predator defense, but not when males produced unspecific alarm calls”

28: Change to “The alarm calls of putty-nosed monkeys were not only sexually dimorphic, but there were also sex-specific differences in alarm call usage and in the responses to calls”. Writing “we discuss” is not very satisfactory. Please try harder to give the reader an idea what you are discussing.

35: As I noted before, the idea that alarm calling is extremely costly is overrated (IMHO). Perhaps tone it down a bit and mention that there could be sex-specific differences in the costs?

39: "heterospecifics" is distracting here; not the topic of the paper

43: The paper you mention does not make any claims about inter-individual variation. It deals with variation between populations.

44: from all that we know from early ontogeny studies but also the more recent paper by Wedgell and colleagues 2019 NEE, there is an evolved disposition to use e.g. an aerial alarm in response to aerial threat. What is learned is what exactly constitutes a danger.

54: → dimorphic

63: rewrite: "if there are also sex-specific differences in ... "

I understand that the predictions were formulated in response to a reviewer's request. The predictions formulated in lines 66 ff are based on results in Diana monkeys. That's not an ideal basis for developing predictions. I would recommend to carefully consider which predictions would be grounded in theory, and otherwise simply say that you explore the behavior.

73: → form a more cohesive cluster

85: Mentioning of individual identity can be deleted – not the issue here. It's only distracting.

87 ff: move to result section.

96: → "driven by differential selective pressures"

The description of the study is really difficult to follow. Perhaps you can come up with a diagram – like a flow chart - that provides an overview over the conditions and the responses (dependent variables)?

100: delete subclause "which most likely ..." – Keep it simple ... "We predicted that after hearing their females' alarm calling, males would approach the females because ..." "We further predicted that they would be calling while approaching in order to ... "

TBH, some of these predictions seem to be formulated after the results were known. Not a healthy research practice.

146: new paragraph here.

155: had, not have

173: were (change to past tense from present perfect, check other instances as well).

184: “Additionally ... “ hard to read

I would just let go of some of the statistical tests. Why a Wilcoxon test for call rate change?

In many instances, the wording could be simplified. For instance: “Additionally, when ... “ → “We examined how often males approached females after hearing the females’ alarm calls. Such instances were classified as “successful recruitment”.

193: → “between call types used”

204: “Once reaching the group ... “ no data are presented. Delete or formulate in a more formal manner. E.g. “when males reached the group ... (N = ...)“

206: brackets can be omitted. Change “at this stage ... “ to “In all of these cases, males ... “ and delete “(in 9 out of 9 cases). Start a new paragraph.

209: I strongly suggest to remove the comparison with the data gathered in 2020 – there is a clear order effect here and the analyses are not particularly meaningful. That is, lines 209-215 could be deleted.

217: the entire section requires an overhaul. The order of results and the mentioning of the corresponding figure should go from 2a to 2f and not start with 2e. By the way, the figures require some more care (e.g. delete default background pattern from ggplot2. I don’t think that 2d and 2f are really needed.

223: Change to “When males did not approach or did not switch to kek calls (correct?), females continued to call”. Too bad that there is no way to de-confound these two aspects.

228: rarely → hardly

231: I don’t understand. I thought that females do not respond. How can you now run an analysis on “before and after female calling”?

236: perhaps simply compare call rates of approaching males vs. males that are not approaching. Note however that this part of the analysis is not terribly informative, as you may be measuring the same thing twice, namely the motivation to engage in anti-predator behavior.

242 ff: I found this part quite confusing. I would recommend to delete everything after “In these cases ... “

262: delete: “during predation events” or change to “simulated predation events”.

272: New paragraph here

280: Delete, there is no evidence

Lines 284 ff – line 317 is beside the point and should be deleted. Instead, the issue of differential vulnerability could be taken up again. The final paragraphs should be more clearly labelled as “outlook”. The issue with sexual selection should be formulated as a direction for future research.

Please check the references carefully. In some instances, species names need to be set in italics. There are also some typos, e.g. 362: groupmembers, and formatting issues.

Appendix B

Dear Dr. Schülke,

hereby we would like to resubmit our manuscript entitled “Female putty-nosed monkeys (*Cercopithecus nictitans*) vocally recruit males for predator defence” (RSOS-200430) to be re-evaluated for publication. We hope that we have dealt with requests and comments to your and the reviewer’s satisfaction.

Amongst others, we rewrote the abstract to focus more on the findings, we restructured the results and rephrased substantial parts of the discussion, and we added a new figure illustrating female and male vocal and behavioural response patterns in different conditions. Please find specific changes copied and current line numbers in the manuscript below.

We are grateful for your effort dealing with our manuscript and any comments/corrections made by the reviewer and by you.

With regards,

Claudia Stephan (on behalf of both authors)

Responses to reviewer

Abstract: Overall, the abstract is poorly structured. The first sentence is about call structure, the second about call usage and responses to calls.

L 14: “Nonetheless, communicative strategies ... “ This statement appears to be a straw man. Why don’t you start with a more general statement that the costs and benefits of alarm calling are expected to vary in relation to individual vulnerability, the presence of kin in the group, and sexual selection. These factors may explain why alarm call patterns and responses to alarm calls may differ between the sexes. Then you go on with “We here experimentally investigate”. The thought outlined above could be unpacked in the introduction section.

16: “another forest guenon” – the reader can only guess what you are trying to say. Delete “another”

19: reassuring □ replace with “ensuring”

22: delete “had”

23: Full stop after “group”, then start new sentence with “They ...(and insert “also” here).

25: Change to “females only ceased to alarm call when males produced calls typically associated with anti-predator defense, but not when males produced unspecific alarm calls”

28: Change to “The alarm calls of putty-nosed monkeys were not only sexually dimorphic, but there were also sex-specific differences in alarm call usage and in the responses to calls”. Writing “we discuss” is not very satisfactory. Please try harder to give the reader an idea what you are discussing.

We have rewritten the entire abstract, taking into account points mentioned above (line 10 – 29):

“Alarm calls can trigger very different behavioural changes in receivers and signallers might apply different alarm call strategies based on their individual cost-benefit ratio. These cost-benefit ratios can also vary as a function of sex. For instance, male but not female forest guenons possess loud alarms that serve warning and predator deterrence functions, but also intergroup spacing and male-male competition. In some forest guenons, the context-specificity and alarm call repertoire size additionally differs between females and males but it remains unclear if this corresponds to similar sexual dimorphisms in alarm calling strategies. We here experimentally investigate whether general female and more context-specific male alarm calls in putty-nosed monkeys (*Cercopithecus nictitans*) had different effects on the opposite sex’ behaviour and whether they might serve different female and male alarm calling strategies. We presented a leopard model separately to the females or to the male of several groups while ensuring that the opposite sex only heard alarm calls of target individuals. While female alarms led to the recruitment of males in the majority of cases, male alarms did not have a similar effect on female behaviour. Males further seem to vocally advertise their engagement in group defence with more unspecific alarms while approaching their group. Males switched alarm call types once they spotted the leopard model and started mobbing behaviour. Females only ceased to alarm call when males produced calls typically associated with anti-predator defence, but not when males produced unspecific alarm calls. Our results suggest that sexual dimorphisms in the context-specificity of alarms most likely correspond to different alarm calling strategies in female and male putty-nosed monkeys.”

35: As I noted before, the idea that alarm calling is extremely costly is overrated (IMHO). Perhaps tone it down a bit and mention that there could be sex-specific differences in the costs?

We have omitted the notion on costly alarm calling and instead refer to different cost-benefit ratios for individual signalers and between sexes (line 48 - 52):

“The cost-benefit ratio that guides signaller behaviour can vary between individuals (e.g. depending on their spatial position and vulnerability, the number of present kin, opportunities for reproduction). If females and males systematically differ in costs-benefit ratios, this might lead to at least partly sex-specific strategies during alarm calling [11]. In these cases, sex-specific differences in alarm calls are expected to facilitate cognitive inferences from calls and behavioural interactions.”

39: “heterospecifics” is distracting here; not the topic of the paper

We have deleted reference to heterospecifics.

43: The paper you mention does not make any claims about inter-individual variation. It deals with variation between populations.

We have deleted reference to call production here.

54: □ *dimorphic*

We have changes the sentence to (line 55 - 57):

“Some forest guenon species possess sexually dimorphic alarm calls with male and female alarms substantially differing in e.g. call morphology [12, 13], repertoire size [14] or in the distance over which their calls carry [15].”

63: *rewrite: “if there are also sex-specific differences in ... “*

I understand that the predictions were formulated in response to a reviewer’s request. The predictions formulated in lines 66 ff are based on results in Diana monkeys. That’s not an ideal basis for developing predictions. I would recommend to carefully consider which predictions would be grounded in theory, and otherwise simply say that you explore the behavior.

We have removed the reference to previous finding in Diana monkeys here.

73: □ *form a more cohesive cluster*

We have rewritten the sentence to (line 76 - 78):

“Males usually socially interact less than females and they are often found at the group’s periphery while females and offspring form a more cohesive cluster [20].”

85: *Mentioning of individual identity can be deleted – not the issue here. It’s only distracting.*

We have opted for leaving the notion of individual identity as this might explain why male putty-nose monkeys start alarm calling with “pyows” during approaches, i.e. giving cues to identity that are linked to anti-predation services. We are referring back to this in the discussion.

87 ff: *move to result section.*

We previously mentioned “kek” calls in the result section and moved it to the introduction following reviewers, who requested to introduce kek calls earlier. We would like to leave the final decision to the editor.

96: □ *“driven by differential selective pressures”*

The description of the study is really difficult to follow. Perhaps you can come up with a diagram – like a flow chart - that provides an overview over the conditions and the responses (dependent variables)?

We simplified the study description (line 97 - 102) and we have added a flowchart illustrating conditions and responses (Fig.2):

“We here experimentally investigated communicative strategies in female and male putty-nosed monkeys during a simulated predatory event. Specifically, we separated the information about an ongoing event that was available to females and males. In the female condition, a leopard model was presented to females and males only heard female alarms. In the male condition, a leopard model was presented to the male of a group and now females only heard male alarms.”

100: delete subclause “which most likely ...” – Keep it simple ... “We predicted that after hearing their females’ alarm calling, males would approach the females because ...” “We further predicted that they would be calling while approaching in order to ...”

TBH, some of these predictions seem to be formulated after the results were known. Not a healthy research practice.

We have rewritten the prediction section (line 102 - 109):

“We explored female and male call usage and behavioural interactions to see if response patterns were sex-specific. We predict that males are not able to infer specific information about the threat by only hearing general female calls and that they should approach and verify the situation themselves in order to adapt anti-predation strategies. For females, we predict no approaches to their calling male because more specific male alarms already provide them with information that allows for specific predation defence. Possible alarm calling strategies and how differences in alarm call specificity could serve different strategies in females and males are discussed.”

146: new paragraph here.

155: had, not have

173: were (change to past tense from present perfect, check other instances as well).

193: □ “between call types used”

We have made respective changes.

184: “Additionally ...” hard to read

I would just let go of some of the statistical tests.

In many instances, the wording could be simplified. For instance: “Additionally, when ...” □

“We examined how often males approached females after hearing the females’ alarm calls.

Such instances were classified as “successful recruitment”.

We omitted the statistical analysis of call rate change for successful and unsuccessful recruitment. We nonetheless mention the difference in call rates for male approaching and males not approaching their group in the results and in Fig.3 to illustrate that call rate was linked to behaviour (approach) and might thus provide clues to male motivation for anti-predation defence.

204: “Once reaching the group ...” no data are presented. Delete or formulate in a more

formal manner. E.g. “when males reached the group ... (N = ...)”

line 197: “When the male reached his group, he scanned the surrounding (N=13).”

206: brackets can be omitted. Change “at this stage ... “ to “In all of these cases, males ... “ and delete “(in 9 out of 9 cases). Start a new paragraph.

line 209/210: “As soon as the male approached its group and detected the leopard model, he always switched from pyow to kek calls (9 of 9 trials).”

209: I strongly suggest to remove the comparison with the data gathered in 2020 – there is a clear order effect here and the analyses are not particularly meaningful. That is, lines 209-215 could be deleted.

We have collected and included these data in response to a former reviewer comment questioning the baseline of spatial approaches between males and females in neutral contexts. We also agree with the present comment that it might disturb the reading flow here. We have moved information provided here to the methods section (line 160ff) and only refer here to comparisons with usual spatial behavior (line 194 - 196):

“In the female condition, when females were confronted with the leopard model and started alarm calling, the male approached the rest of the group in 13 of 17 cases, which is more often than would be expected by usual male and female spatial behaviour (binominal test, $p=0.041$).”

217: the entire section requires an overhaul. The order of results and the mentioning of the corresponding figure should go from 2a to 2f and not start with 2e. By the way, the figures require some more care (e.g. delete default background pattern from ggplot2. I don't think that 2d and 2f are really needed.

228: rarely □ hardly

231: I don't understand. I thought that females do not respond. How can you now run an analysis on “before and after female calling”?

236: perhaps simply compare call rates of approaching males vs. males that are not approaching.

242 ff: I found this part quite confusing. I would recommend to delete everything after “In these cases ... “

We have restructured and rephrased the entire section and removed default backgrounds from the figures (line 206 - 225, Fig.3):

“In the female condition, when the male did not see the leopard model but only received female alarms, they were always triggered to emit pyow alarms (N=17). When hearing their male's pyow calls, females did not change their alarm calling (before and after male pyows: N=17, W=81, $p=0.586$, Fig. 3a). As soon as the male approached its group and detected the leopard model, he always switched from pyow to kek calls (9 of 9 trials). In response, females ceased own alarm

calling (before and after first male kek: Wilcoxon signed rank test, $N=9$, $W=55$, $p=0.003$, Fig. 3b). When the male did not approach females or he approached but did not switch to anti-predation defence (8 of 17 trials), the male also did not switch to kek calls and females persisted in alarm calling (female response duration: Mann-Whitney U-test, $N_{\text{no_kek}}=8$, $\text{median}_{\text{no_kek}}=575$ sec, $N_{\text{kek}}=9$, $\text{median}_{\text{kek}}=174$ sec, $U=8$, $p=0.003$). Males called overall more when they approached their group than when they did not approach their group (successful recruitments: 0.46 calls/sec after female alarms, Fig. 3c; unsuccessful recruitments: 0.1 calls/sec after female alarms, Fig. 3d).

In the male condition, when males saw the leopard model first and females were oblivious to the nature of the threat, males uttered kek calls without preceding pyows and females hardly responded at all (before and after the onset of male responses: Wilcoxon signed rank tests, $N=9$, $W=0$, $p=0.99$; Fig. 3e). Males did not change their call rates towards the leopard model after females started their weak alarm response (before and after the onset of female responses: $N=9$, $W=39.5$, $p=0.96$, Fig. 3f)."

262: *delete: "during predation events" or change to "simulated predation events"*.

Line 237 - 239: "Our results show that female and male putty-nosed monkeys monitor each other's alarm calling and that their sex-specific alarm calls mediate behavioural changes in the opposite sex during simulated predation events."

272: *New paragraph here*

We have rewritten this part of the discussion.

280: *Delete, there is no evidence*

We have deleted the sentence.

Lines 284 ff – line 317 is beside the point and should be deleted. Instead, the issue of differential vulnerability could be taken up again. The final paragraphs should be more clearly labelled as "outlook". The issue with sexual selection should be formulated as a direction for future research.

We deleted the entire section and replaced it by a summary paragraph including future research topics (line 276 – 289):

"In sum, results presented here show that female and male putty-nosed monkey alarm calls follow different strategies when alarm calling that can be partly explained by individual and kin selection. However, individual and kin selection do not straightforwardly explain why males possess different call types and why they start calling during approaches without having information about the threat. The fact that pyow calls provide females with information about the signaller's identity [24] could be important for group-holding males' reputation as reliable group

defenders and their tenure track length [29], especially in dense forest habitats where visibility is usually low. It yet remains unknown to what degree female forest guenons have control over mate choice as seen in other non-human primates forming multi-male groups [19, 30, 31]. One specific hypothesis would be that males, who do not reliably take over anti-predation defence also have less reproductive success, by e.g. shorter tenure duration, than more reliable callers and group defenders. The investigation of male reliability in group defence as indicated by alarm call usage, and female mate preference provide promising ground for future research.”

Please check the references carefully. In some instances, species names need to be set in italics. There are also some typos, e.g. 362: groupmembers, and formatting issues.

We checked references and made respective changes.